# Mineralization Profile of Annexin A6-Harbouring Proteoliposomes: Shedding Light on the Role of Annexin A6 on Matrix Vesicle-Mediated Mineralization

**DOI:** 10.3390/ijms23168945

**Published:** 2022-08-11

**Authors:** Ekeveliny Amabile Veschi, Maytê Bolean, Luiz Henrique da Silva Andrilli, Heitor Gobbi Sebinelli, Agnieszka Strzelecka-Kiliszek, Joanna Bandorowicz-Pikula, Slawomir Pikula, Thierry Granjon, Saida Mebarek, David Magne, José Luis Millán, Ana Paula Ramos, Rene Buchet, Massimo Bottini, Pietro Ciancaglini

**Affiliations:** 1Departamento de Química, Faculdade de Filosofia, Ciências e Letras de Ribeirão Preto da Universidade de São Paulo (FFCLRP-USP), Ribeirão Preto 14040-901, SP, Brazil; 2Nencki Institute of Experimental Biology, 3 Pasteur Street, 02-093 Warsaw, Poland; 3University of Lyon, University Claude Bernard Lyon 1, CNRS UMR 5246, ICBMS, F-69622 Lyon, France; 4Sanford Burnham Prebys, La Jolla, CA 92037, USA; 5Department of Experimental Medicine, University of Rome Tor Vergata, 00133 Rome, Italy

**Keywords:** annexin A6, proteoliposome, biomineralization, apatite, matrix vesicle mimetic model

## Abstract

The biochemical machinery involved in matrix vesicles-mediated bone mineralization involves a specific set of lipids, enzymes, and proteins. Annexins, among their many functions, have been described as responsible for the formation and stabilization of the matrix vesicles′ nucleational core. However, the specific role of each member of the annexin family, especially in the presence of type-I collagen, remains to be clarified. To address this issue, in vitro mineralization was carried out using AnxA6 (in solution or associated to the proteoliposomes) in the presence or in the absence of type-I collagen, incubated with either amorphous calcium phosphate (ACP) or a phosphatidylserine-calcium phosphate complex (PS–CPLX) as nucleators. Proteoliposomes were composed of 1,2-dipalmitoylphosphatidylcholine (DPPC), 1,2-dipalmitoylphosphatidylcholine: 1,2-dipalmitoylphosphatidylserine (DPPC:DPPS), and DPPC:Cholesterol:DPPS to mimic the outer and the inner leaflet of the matrix vesicles membrane as well as to investigate the effect of the membrane fluidity. Kinetic parameters of mineralization were calculated from time-dependent turbidity curves of free Annexin A6 (AnxA6) and AnxA6-containing proteoliposomes dispersed in synthetic cartilage lymph. The chemical composition of the minerals formed was investigated by Fourier transform infrared spectroscopy (FTIR). Free AnxA6 and AnxA6-proteoliposomes in the presence of ACP were not able to propagate mineralization; however, poorly crystalline calcium phosphates were formed in the presence of PS–CPLX, supporting the role of annexin-calcium-phosphatidylserine complex in the formation and stabilization of the matrix vesicles’ nucleational core. We found that AnxA6 lacks nucleation propagation capacity when incorporated into liposomes in the presence of PS–CPLX and type-I collagen. This suggests that AnxA6 may interact either with phospholipids, forming a nucleational core, or with type-I collagen, albeit less efficiently, to induce the nucleation process.

## 1. Introduction

Mineralization of soft tissues is an orchestrated biochemical process aimed at balancing the concentration of the potent calcification inhibitor inorganic pyrophosphate (PP_i_) and the local concentrations of phosphate (P_i_), a calcification promoter, to establish a proper PP_i_/P_i_ ratio to enable calcification to initiate and propagate [1,2,3,4,5,6]. The formation of apatite can be induced by a special class of medium-sized extracellular vesicles, named matrix vesicles (MVs) [7,8,9,10,11,12]. MVs are released from hypertrophic chondrocytes and mature osteoblasts, the cells responsible for endochondral and intramembranous ossification, respectively [7,8,9,10,11,12]. The MV-driven mineralization goes through a multi-step process: MVs are released by outward budding from the apical microvilli of mineralizing cells, bind to the collagenous matrix, and establish the proper PP_i_/P_i_ ratio to initiate the growth of apatite in their lumen [2,3,4,5,6,7]. Then, apatite is released in the extracellular milieu and propagates onto the collagen fibrils. MVs intraluminally accumulate P_i_ generated locally by the action of specific enzymes, including PHOSPHO1, which hydrolyses phosphocholine and phosphoethanolamine [13,14], and neutral sphingomyelinase 2 (encoded by sphingomyelin phosphodiesterase 3 gene SMPD3) [15], which forms phosphocholine from sphingomyelin. In addition, P_i_ is generated extracellularly by tissue-nonspecific alkaline phosphatase (TNAP [2]) from the hydrolysis of phosphoesters as well as by ectonucleotide pyrophosphatase/phosphodiesterase 1 (NPP1) from the hydrolysis of extracellular ATP [4] and incorporated in the MVs’ lumen via specific sodium-dependent (e.g., P_i_T-1) [16,17,18] or sodium-independent [19] transporters. How MVs accumulate calcium in their lumen has not yet been clarified. Voltage-dependent L-type Ca^2+^ channels are present in the early stage of growth plate development [10]. However, MVs lack the voltage differential to drive Ca^2+^ uptake [10]. Several members of the annexin family (AnxA2, AnxA5, and AnxA6) can be found in the proteomic profile of MVs and may function as ion channels [10,20]. Unidentified factors or other types of channels may contribute to the Ca^2+^ uptake by MVs [10,21]. It has been proposed that AnxA2, AnxA5, and AnxA6 may be also involved in the formation of a nucleational core associated with phosphatidylserine in the MVs’ lumen [22]. AnxA6 is thought to tightly bind or insert into the MVs’ lipid bilayer, since a part of the protein cannot be extracted by calcium chelators (e.g., Ethylene glycol-bis(2-aminoethylether)-N,N,N′,N′-tetraacetic acid (EGTA) or Ethylenediaminetetraacetic acid (EDTA)) [20,23,24]. One part corresponds to Ca^2+^-bound AnxA6 that interacts with the inner leaflet of the MVs’ membrane [23,24], a second part corresponds to AnxA6 localized on the surface of the outer leaflet and, lastly, a third part corresponds to AnxA6 inserted in the hydrophobic membrane bilayer and co-localizes with cholesterol [23,24]. AnxA6 has been found to be less efficient than AnxA5 in initiating nucleation in vitro [22]. Since MVs are strongly bound to the extracellular matrix, and especially to collagen [20], there is the possibility that collagen may affect the nucleation process.

In the current report, we assess how collagen affects the role of AnxA6 on MV-mediated mineralization by using proteoliposomes. The ability of proteoliposomes to mimic the function and structure of biomembranes makes them a convenient model to assess the mechanisms of MV-mediated mineralization [22,25,26]. We used proteoliposomes harbouring AnxA6 and composed by either 1,2-dipalmitoylphosphatidylcholine (DPPC) to mimic the outer leaflet, or a mixture of 1,2-DPPC and 1,2-dipalmitoylphosphatidylserine (DPPS) to mimic the inner leaflet, or a mixture of DPPC, DPPS, and cholesterol (Chol). The ability of the proteoliposomes to mineralize was assessed in the presence of each component of the MVs’ nucleational core, that is, amorphous calcium phosphate or phosphatidylserine-complex, and in the presence or in the absence of type-I collagen.

## 2. Results and Discussion

### 2.1. Biophysical Properties of Liposomes and Proteoliposomes

Proteoliposomes were fabricated by incubating AnxA6 with liposomes made of either 1,2-dipalmitoylphosphatidylcholine (DPPC proteoliposomes), or a mixture of 1,2-dipalmitoylphosphatidylcholine and 1,2-dipalmitoylphosphatidylserine with a 9:1 molar ratio (DPPC:DPPS (9:1) proteoliposomes), or 1,2-dipalmitoylphosphatidylcholine, cholesterol and 1,2-dipalmitoylphosphatidylserine with a 5:4:1 molar ratio (DPPC:Chol:DPPS (5:4:1) proteoliposomes) by following protocols previously optimized by our research group [23]. The diameter of the liposomes was evaluated by DLS. All obtained vesicles had an average diameter between 114 and 199 nm with a polydispersion index (PI) smaller than 0.09, suggesting the formation of monodisperse vesicles (Table 1). The incorporation of 17 µg/mL AnxA6 was achieved for all the three lipid mixtures used, as described by Veschi et al. [23]. The proteoliposomes’ diameter did not differ significantly from that of the corresponding liposomes (Table 1) [23].

### 2.2. Biomineralization Mediated by AnxA6-Harbouring Proteoliposomes with Amorphous Calcium Phosphate

AnxA6 without ACP did not mineralize in any of the three lipid mixtures tested. The ability of the three types of AnxA6-harboring proteoliposomes (DPPC, DPPC:DPPS (9:1), DPPC:Chol:DPPS (5:4:1)) to propagate mineralization in synthetic cartilage lymph (SCL) in the presence of amorphous calcium phosphate (ACP), with and without type-I collagen, was assessed by means of turbidimetry. AnxA6 alone was found to be insufficient to induce mineral formation in SCL medium (Figure 1, green curve). We also found that none of the tested proteoliposomes in the absence of cholesterol and type-I collagen mineralized when ACP was used as the nucleator, as shown in (Figure 1A, blue and orange curves). A slight monotone increase in the turbidity was observed after incubation with the DPPC:Chol:DPPS (5:4:1) proteoliposome (Figure 1A, black curve), which was interpreted as a random stochastic linear increase in the turbidity and not a nucleation process. The addition of type-I collagen to each of the three types of proteoliposomes induced a monotone increase in the turbidity, which was also observed in the case of free AnxA6 in solution (Figure 1B), suggesting that the metastable SCL medium containing type-I collagen formed aggregates in all of the conditions tested.

### 2.3. Biomineralization Mediated by AnxA6-Harbouring Proteoliposomes with Phosphatidylserine-Calcium Complex in the Absence of Type-I Collagen

The ability of the AnxA6-harbouring proteoliposomes (DPPC, DPPC:DPPS (9:1), DPPC:Chol:DPPS (5:4:1)) to mineralize in the presence of PS–CPLX was investigated both in the absence (Figure 2A) and in the presence (Figure 2B) of type-I collagen, in SCL medium. AnxA6 alone with PS–CPLX induced a slight increase in the turbidity in the first 20 h of contact, which might be related to a reminiscent nucleation process. The turbidity increased after 25–30 h incubation, which was followed by its decrease, suggesting the formation of calcium phosphate complexes associated with AnxA6 and PS–CPLX, which then precipitated (Figure 2A, green curve).

When AnxA6 was mixed with PS–CPLX and each of the liposomes (DPPC, DPPC:DPPS (9:1), or DPPC:Chol:DPPS (5:4:1)), two simultaneous processes could be identified: (1) a random stochastic or monotonic process that showed a characteristic linear increase in turbidity as a function of time; and (2) a nucleation step with a sigmoidal increase in turbidity as a function of time, mediated by the set of essential components for the formation of minerals (Figure 2A, black, orange, and blue curves). The occurrence of these two processes might be related to the experimental conditions, that is, the presence of 1.41 mM Pi and 2 mM Ca^2+^ in the synthetic cartilage medium, sufficient to trigger mineral precipitation. The linear increase reflects a steady stochastic precipitation process. Since nucleation is a synergic process with a fast formation of nuclei, it results in a characteristic sigmoidal curve with a sharp transition. Sigmoidal curves describing the nucleation process in synthetic cartilage medium were previously reported by Genge et al. [22]. 

To obtain the nucleation curves, the monotonic step was subtracted from the turbidity curves in the case of AnxA6 mixed with PS–CPLX and each of the liposomes (DPPC, DPPC:DPPS (9:1), or DPPC:Chol:DPPS (5:4:1)) (Appendix A).

Then the nucleation curves were fitted by a sigmoidal function to determine the kinetic parameters, including the initial time of mineral formation (t_i_), the final time of mineral formation (t_f_), the time at maximum rate of mineral formation (t_max rate_), the maximum value of turbidity (U_max_), and the potential of mineral propagation (PMP = U_max_/t_max rate_) [22]. In the absence of type-I collagen, the nucleation induced by DPPC proteoliposomes started approximately 10–15 h after the addition of PS–CPLX. A t_max rate_ was achieved after approximately 18.5 h, in which a U_max_ of 0.19 with a PMP of 0.010 h^−1^ was reached (Appendix A and Table 2). DPPC:DPPS (9:1) proteoliposomes showed a nucleation profile with kinetic parameters slightly different (t_Max rate_ = 22.9 h; U_max_ = 0.15; and PMP = 0.007 h^−1^) to that of DPPC proteoliposomes (Appendix A and Table 2). DPPC:Chol:DPPS (5:4:1) proteoliposomes exhibited a nucleation curve with higher values of U_max_ = 0.22 and PMP = 0.011 h^−1^ than the other proteoliposomes. Finally, AnxA6 with PS–CPLX alone showed a t_i_ = 20.7 h, t_max rate_ = 22.4 h, U_max_ = 0.42, and PMP = 0.019 h^−1^ (Figure 2A, green curve and Table 2). Since the monotonic increase in the turbidity was absent in the case of AnxA6 with PS–CPLX alone, its nucleation process has the greatest magnitude U_max_ = 0.42 as compared to the other nucleation process induced with the lipids.

The addition of type I collagen almost completely hampered the mineral propagation induced by AnxA6 and PS–CPLX (Figure 2B, green curve). This may suggest that the interaction between AnxA6 and PS–CPLX was abolished due to the binding between PS–CPLX and type-I collagen. On the other hand, we observed a sharp increase in the turbidity at around 20–25 h incubation, reminiscent of a nucleation process, which was induced by each of the three types of AnxA6 harbouring proteoliposomes with PS–CPLX in the presence of type-I collagen (Figure 2B, black, orange, and blue curves). In the presence of type-I collagen, the proteoliposomes showed similar mineralization curves, irrespective of the lipid composition (Figure 2B), which also translated to very similar values of PMP (0.016 to 0.018 h^−1^) (Table 3). AnxA6 in the absence of type-I collagen indicated a similar PMP as AnxA6 harbouring proteoliposomes in the presence of type-I collagen (Table 2). The t_i_ of AnxA6 harbouring proteoliposomes in the presence of type-I collagen was in the range of 13.8 to 15.2 h (Table 3). Additionally, the values of potential of mineral propagation (PMP) in the presence of type I collagen were higher than those in the absence of collagen for all the tested vesicles (Table 2 and Table 3). Since physiological mineralization takes place in vivo and in the collagen-enriched matrix, these results suggest the main role of AnxA6 in the initial steps of mineralization, specifically the formation of the nucleation core occurs before the MVs binding to the extracellular matrix.

Depending upon the conditions of temperature, pH, and ion concentrations, the calcium and phosphate ion product will yield a variety of forms, including amorphous calcium phosphate (ACP), brushite [CaHPO_4_·2H_2_O], whitlockite [Ca_9−x_(MgFe)_x_(PO_4_)_6_PO_3_OH], octacalcium phosphate [Ca_8_H_2_(PO_4_)_6_·5H_2_O], and hydroxyapatite [Ca_10_(PO_4_)_6_(OH)_2_] [27,28,29,30]. Biological apatite has a distinct composition from hydroxyapatite and may contain Cl^−^, F^−^, CO_3_^2−^, HPO_4_^−^, as well as Sr^2+^, Ba^2+^, Mg^2+^, Zn^2+^, Na^+^, and K^+^, and present with lacunae in the crystal lattice [31,32,33]. ACP formation is thought to proceed through prenucleation clusters that are present in the solution before nucleation, as reported for amorphous calcium carbonates (ACC) [34]. A nucleation mechanism is necessary to initiate apatite formation in an efficient manner [35]. In this respect MVs can initiate apatite formation in their lumen [7,8,9,10,11,12]. It was proposed that AnxA2, AnxA5, and AnxA6 could form a nucleational core with PS in the MVs’ lumen [22]. Indeed, our findings showed that AnxA6 can propagate the mineral more efficiently when the PS–CPLX complex was used as a nucleator compared to ACP for all the types of liposomes tested. It seems that there was no significant nucleation mechanism induced by the addition of ACP, as indicated by the monotone increase in turbidity (Figure 1), which could correspond to a stochastic mineralization process. The affinity of annexins to PS and their key role for the formation of an annexin-PS-mineral complex that stabilizes the nucleational core inside the MVs core has been proposed by Wuthier [36]. Our results attest the affinity of Anx-A6 to PS and its key role during the propagation of mineralization, as also previously shown by Veschi et al. [23].

Based on the turbidity at 340 nm, and as standardized by Wu et al. [37], we estimated the percentage of Ca^2+^ precipitated in the presence of AnxA6 in solution and bound to the proteoliposomes in the presence of PS–CPLX. An increase of 0.1 in turbidity is equivalent to the precipitation of approximately 10% of the total Ca^2+^ in SCL [22,37]. The percentage of Ca^2+^ precipitated was ~40% for AnxA6 in solution in the absence of type-I collagen, while it was 5% for AnxA6 in solution in the presence of type-I collagen. All the proteoliposomes showed the percentage of Ca^2+^ precipitated ranging from 35% to 55% in the absence of type-I collagen and of approximately 45–55% in the presence of collagen. Collagen fibrils can induce the deposition of apatite crystallites through a heterogeneous nucleation process and interfibrillar non-collagenous proteins are thought to be co-factors that permit crystallite deposition [38].

### 2.4. Spectroscopic Analysis of the Minerals Formed by Annexin A6-Harbouring Proteoliposomes

An FTIR analysis of the minerals precipitated in the presence of the proteoliposomes was carried out in order to support the different kinetic profiles found in the turbidimetry analysis. The samples were precipitated in the presence of CPLX as a nucleator, since the amount of minerals formed in the presence of ACP was not enough to be analysed. Moreover, since it is not possible to discard the contribution of collagen to the bands, especially in the 1500–1000 cm^−1^ range, we analysed only the samples precipitated in absence of the protein.

MVs are thought to induce a nucleation process within 2–4 h that yields apatite [39]. The proteoliposomes harbouring AnxA6, in the presence of ACP, showed a monotone increase in the turbidity (Figure 1) and did not induce a nucleation process to yield significant amount of minerals to be analysed. In the case of proteoliposomes harbouring with PS–CPLX in the absence of collagen, there was a superposition of a monotonous increase in the turbidity and a nucleation process which was observed after 10–15 h (Appendix A), while free AnxA6 in the absence of the lipids and in the presence of PS–CPLX induced only a nucleation process which appeared after 20 h.

The minerals produced by each proteoliposome harbouring AnxA6 as well as by free AnxA6 in the presence of PS–CPLX were assessed by the FTIR analysis (Figure 3).

Four groups of bands were observed in all the FTIR spectra (Figure 3): (1) Two broad bands in the range of 1150 to 1090 cm^−1^, assigned to the formation of poorly crystalline calcium phosphates [40] and/or phosphate in acidic environment [41]; (2) A peak centred at 1040–1030 cm^−1^, assigned to the asymmetric stretching of the PO_4_^3−^ group which may correspond to apatite [41,42,43,44,45,46]; (3) A peak centred at 985 cm^−1^ which was assigned to the HPO_4_^2−^ group [46], which may indicate brushite; and (4) A small band at 925 cm^−1^ assigned to the P-O-P group [46]. The spectral profiles suggested the formation of poorly crystalline calcium phosphates irrespective of the composition of the proteoliposomes, in which the presence of acidic (HPO_4_^2−^) phases and apatite (PO_4_^3−^) can be identified. The relative intensity and position of the bands ranging from 1130 to 1030 cm^−1^ served to analyze the maturation of the mineral precipitated from MVs incubated into SCL [47]. The ratio between the intensity of the bands at 1040–1030 and 985 cm^−1^ may help to discriminate the composition of the samples precipitated by the proteoliposomes tested.

The formation of apatite was slightly favoured in the presence of AnxA6-DPPC proteoliposomes, compared to the other proteoliposomes and AnxA6 in solution (Table 4). This was consistent with the fact that the formation of mature apatite phases can be induced by TNAP-pure DPPC proteoliposomes and ATP as a TNAP substrate [48]. Our findings show that although the formation of a more apatitic mineral phase was favoured in the presence of AnxA6-DPPC proteoliposomes, the presence of PS–CPLX may stabilize poor crystalline and acidic phases.

### 2.5. Circular Dichroism of Annexin A6 in Different pHs

The presence of acidic phosphate as indicated by the IR bands at around 1150–1090 cm^−1^ and at 985 cm^−1^ (Figure 3) may suggest a slight acidification of the SCL buffer due to the formation of apatite. Such acidification during apatite formation induced by MVs was reported [24]. The pH affected the Ca/P atomic ratio (1.42 at pH 5.8, and 1.48 at pH 6.5) while their XRD and IR spectra remained similar at both pHs [49]. AnxA6 at pH 7.4 interacted on the surface of DMPC:DPPS (9:1), while it increased the fluidity of the DMPC:DPPS (9:1) proteoliposome at pH 5.4, suggesting its insertion into the bilayer [24]. The interactions between AnxA6 and lipids can be affected by a pH variation.

AnxA6 exhibits the highest percentage of α-helix structures at pH 6.5 (86%) (Figure 4), which agrees with a previous study [50]. The pH variations are more effective in affecting the function and structure of annexins than variations in Ca^2+^ concentrations [50]. AnxA6 is often detected on the surface of low internal pH cell organelles [51].

## 3. Material and Methods

### 3.1. Materials

All aqueous solutions were prepared using Millipore^®^ DirectQ ultrapure water. Bovine serum albumin (BSA), tris hydroxymethyl-amino-methane (Tris), sodium dodecylsulfate (SDS), and Type I collagen were obtained from Sigma Aldrich (Sigma Aldrich Corp., St. Louis, MO, USA). Bradford reagent was from Bio-Rad, (Bio-Rad Laboratories, Inc., Hercules, CA, USA). DPPC, DPPS and Chol were purchased from Avanti Polar Lipids (Avanti Polar Lipids, Inc., Alabaster, AL, USA), whereas DPPS was obtained from Sigma Aldrich. All reagents were of analytical grade and used without further purification.

### 3.2. Expression of Annexin A6

Recombinant human AnxA6 protein was expressed and purified as described by Bandorowicz-Pikula et al. [52].

### 3.3. Preparation of Liposomes

Liposomes made of either DPPC, or a mixture of DPPC and DPPS with a 9:1 molar ratio, or DPPC, cholesterol (Chol), and DPPS with a 5:4:1 molar ratio were prepared by incubating the lipids in 50 mM Tris-HCl buffer, pH 7.5, with a total concentration of 10 mg·mL^−1^, as previously described by us [23,53].

### 3.4. Preparation of Proteoliposomes

AnxA6 (0.2 mg·mL^−1^) was incorporated into liposomes by direct insertion into a 50 mM Tris-HCl buffer, pH 7.5, containing 2 mM MgCl_2_ in a 1:100 protein:lipid ratio. The mixture was incubated for 24 h at 25 °C and ultracentrifuged at 100,000× *g* for 1 h at 4 °C. The pellet containing proteoliposomes was resuspended in the same buffer [53]. Protein concentration was measured in presence of 0.2 g·mL^−1^ sodium dodecyl sulfate [54]. Bovine serum albumin was used as a standard.

The sizes of liposomes and proteoliposomes were determined by dynamic light scattering [55] using an N5 submicron particle size analyser (Beckman Coulter Inc., Fullerton, CA, USA). The value of the mean diameter was determined as the center of the size distribution peak. The average value (n = 5) of the liposomes′ diameters was obtained at 25 °C by unimodal distribution. The samples were filtered (0.8 μm) before the analysis.

### 3.5. Synthesis of Amorphous Calcium Phosphate and of Phosphatidylserine Calcium Complex Nucleators

Phosphatidylserine-calcium-complex (PS–CPLX) and amorphous calcium phosphate (ACP) nucleators were synthesized as previously described [22,56].

### 3.6. Mineralization Assays with Proteoliposomes and Characterization by Infrared Spectroscopy

AnxA6-containing liposomes (proteoliposomes) were incubated in a synthetic cartilage lymph (SCL) buffer in the presence of an amorphous calcium phosphate (ACP) or phosphatidylserine-complex (PS–CPLX) at pH 7.5, and in the presence or in the absence of type-I collagen. Synthetic cartilage lymph contained 2 mM Ca^2+^, 1.42 mM NaH_2_PO_4_·H_2_O, 104.5 mM Na^+^, 133.5 mM Cl^−^, 63.5 mM sucrose, 16.5 mM Tris, 12.7 mM K^+^, 5.55 mM glucose, 1.83 mM NaHCO_3_, and 0.57 mM MgSO_4_ [36]. The liposomes (protein-free) as well as AnxA6, alone in the solution, were used as controls. Mineral precipitation/propagation was measured by turbidity (U) at 340 nm using a multi-well microplate assay [37]. Quadruplicate samples (100 μL) were poured into the wells of a 96 well microplate. Turbidity measurements were made after 10 s of agitation followed by 30 h of incubation at 25 °C by using a microplate reader (model SpectraMax^®^ M3, Molecular Devices LLC, San Jose, CA, USA). The results were normalized, subtracting the absorbance value obtained during the first measurement of each proteoliposome. Protein-free liposomes dispersed in SCL were used as the control. The turbidity versus time curves obtained after the incubation of the proteoliposome in SCL presented a characteristic sigmoidal shape. By using these curves, a mathematical approach described by Genge et al. [22] was applied for the determination of mineralization-related parameters: the initial mineralization time (t_i_) is characterized by a rapid increase in U; the final mineralization time (t_f_) is characterized by a decrease in U; the time in which the maximum rate of mineral formation is reached (t_max rate_) corresponds to the maximum of the dU/dt curve; U_max_ is the maximum of turbidity at 340 nm; U_max_/t_max rate_ is the potential of mineral propagation (PMP) that is a measure of the tendency to form mineral [22]. The chemical composition of the precipitates was investigated by means of Fourier transformed infrared spectroscopy (FTIR), using an attenuated total reflectance accessory (germanium crystal with acquisition from 4000 to 600 cm^−1^) model IRPrestige-21, Shimadzu Co., Tokyo, Japan.

### 3.7. Collagen Matrix Coating for Mineralization Assay

Type-I collagen was used in order to produce the collagen matrix coating as described by Bolean et al. [57]. First, collagen was dissolved in 50 mM acetic acid at 1.0 mg/mL and placed under stirring for at least 1 h. After that, the collagen solution was diluted to a final concentration of 125 μg/mL in 50 mM acetic acid. 50 μL of type-I collagen solution and 200 μL of stock buffer were then added to each well in a 96 well plate, corresponding to 6.25 μg collagen per well. To achieve maximal refolding during coating, these pepsin-solubilized collagens were renatured in a neutral buffer during the coating process. The plate was kept covered overnight at 4 °C. The next day, the plate was emptied and blocked with stock buffer containing 1% BSA for blocking non-occupied sites (250 μL/well) for 1 h at room temperature. The plate was washed three times with stock buffer without BSA and used as such to mineralization assays described above, when in the presence of type-I collagen.

### 3.8. Circular Dichroism Spectroscopy

Circular dichroism spectra were recorded with a Jasco 810 spectrophotopolarimeter equipped (Hachioji-shi, Japan) with a Peltier thermostatic system under constant nitrogen flux at 25 °C and with a 0.1 cm quartz cuvette. The protein concentration was in the range of 0.44 mg·mL^−1^ at different pH values. The circular dichroism spectrum of each sample was measured three times at a scan rate of 50 nm.min^−1^.

The average of these scans gave the raw protein spectrum, from which a baseline spectrum, measured using the same cell and water minus the protein component, was subtracted. The signal was recorded as molar ellipticity or residue ellipticity [θ] (deg), which is defined as:[θ] = θ_obs_ × *M*_W_ (or MRW)/10 × d × C
where λ = wavelength; θ_obs_ = observed ellipticity in degrees; *M*w = molecular weight; MRW = mean of residue weight of the repeating unit (the MRW of annexin A6 is 120); d = path length in centimetres; and C = protein concentration in g·mL^−1^ [58]. The amount of α-helix structure (ƒ_H_) of annexin A6 was estimated from the ellipticity per residue at 222 nm (θ_222_) determined by the circular dichroism spectra in each condition by the equation [59]:(ƒ_H_) = −([θ]_222_ + 2340)/30,300

### 3.9. Statistical Analysis

Kinetics and mineralization data are reported as the mean ± SD of triplicate measurements of three different proteoliposome preparations, which was considered to be statistically significant at *p* ≤ 0.05 or *p* ≤ 0.001, as indicated.

## 4. Conclusions

Proteoliposomes harbouring AnxA6 are less efficient than MVs [39] to induce a nucleation core and apatite as indicated by a longer initiation time (10–15 h versus 2–4 h) and a larger mixture of amorphous calcium complexes as compared with that in MVs. Moreover, AnxA6 was less capable than AnxA5 to activate the nucleation process [22]. Taken together this may suggest that AnxA6 has other functions. AnxA6, similar to other members of the annexin family, may recruit specific proteins as well as lipids and participate in the reorganization of membrane domains [60]. In the presence of Ca^2+^, AnxA6 binds to phosphatidylserine (PS) on the inner leaflet of the MVs’ lipid bilayer and could contribute to the formation of a nucleational nucleus, possibly with the aid of AnxA5 [34] (Figure 5A,B). AnxA6 might locally accumulate Ca^2+^ and stabilize Ca^2+^ binding PS, promoting a favourable environment for the formation of apatite (Figure 5C), albeit in a less effective manner than AnxA5 [22]. Followed by a local drop in pH during apatite formation, the insertion of hydrophobic AnxA6 into the matrix vesicles’ bilayer can be promoted (Figure 5D) [24]. Although AnxA6 may have univalent K^+^ conductance [52], it is unclear if the transmembrane AnxA6 has the capacity to transport Ca^2+^ [23] or if another unidentified channel has this ability. The outer pH is neutral, rendering AnxA6 more hydrophilic, which can expel AnxA6 towards the extracellular medium, where AnxA6 may bind to outside-leaflet phosphatidylcholine (PC) [24] (Figure 5E). AnxA6 could associate with AnxA5 on the vesicles′ surface and contribute to reinforce the interactions of MVs with collagen fibers (Figure 5F). This hypothesis is supported by the facts that AnxA5 has affinity for different types of collagen fibres [57] and that AnxA6 may have a competitive interaction either with PS–CPLX or with collagen, since AnxA6 alone with PS–CPLX and collagen did not induce the nucleation process.

Membrane models, such as proteoliposomes and Langmuir monolayers bearing specific proteins, have been useful to investigate the mechanisms of apatite formation and the role of several proteins/enzymes in biomineralization [12,23,26,48,53,55,56,57,61,62,63]. The use of mimetic models of MVs combined with engineered proteins (AnxA6, AnxA5, TNAP, NPP1, PiT1/2, etc.) could positively impact translational medicine through the development of proteoliposomes as nano reactors of mineral formation for the treatment of bone-related diseases. Moreover, these findings provide information about the mineralization capacity of MVs with molecular details. This will ultimately reinforce the current paradigm regarding the biochemical mechanisms that regulate the role of MVs in physiological and pathological calcification.

## Figures and Tables

**Figure 1 ijms-23-08945-f001:**
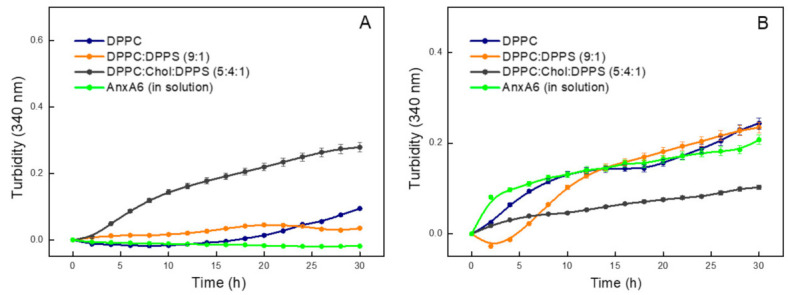
Mineralization curves of proteoliposomes harboring AnxA6 with ACP nucleator. (**A**) Absence of type-I collagen; (**B**) Presence of type-I collagen. Change with time of the value of absorbance at 340 nm of different compositions of proteoliposomes harboring AnxA6. In blue—DPPC; orange—9:1 DPPC:DPPS; black—5:4:1 DPPC:Chol:DPPS; and green—control: AnxA6 in solution.

**Figure 2 ijms-23-08945-f002:**
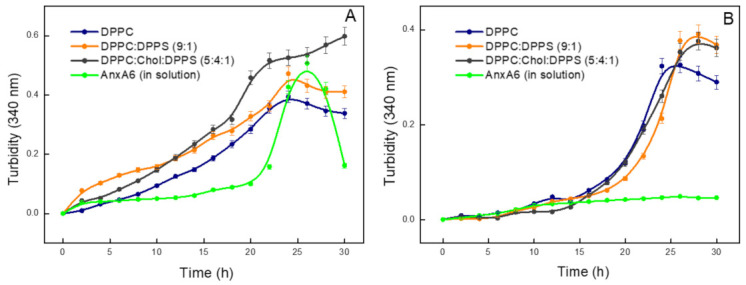
Mineralization curves of proteoliposomes harboring AnxA6 with PS–CPLX nucleator. (**A**) Absence of type-I collagen; (**B**) presence of type-I collagen. Change with time of the value of absorbance at 340 nm of different compositions of proteoliposomes harboring AnxA6. Blue—DPPC; Orange—9:1 DPPC:DPPS; Black—5:4:1 DPPC:Chol:DPPS; and Green—Control: AnxA6 in solution.

**Figure 3 ijms-23-08945-f003:**
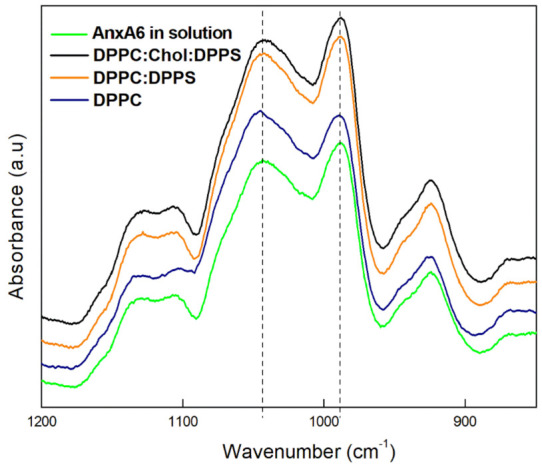
ATR—FTIR spectra of the minerals produced by proteoliposomes harboring AnxA6 after 30 h of incubation in SCL in absence of type-I collagen. Blue—DPPC; Orange—9:1 DPPC:DPPS; Black—5:4:1 DPPC:Chol:DPPS; and Green—Control: AnxA6 in solution.

**Figure 4 ijms-23-08945-f004:**
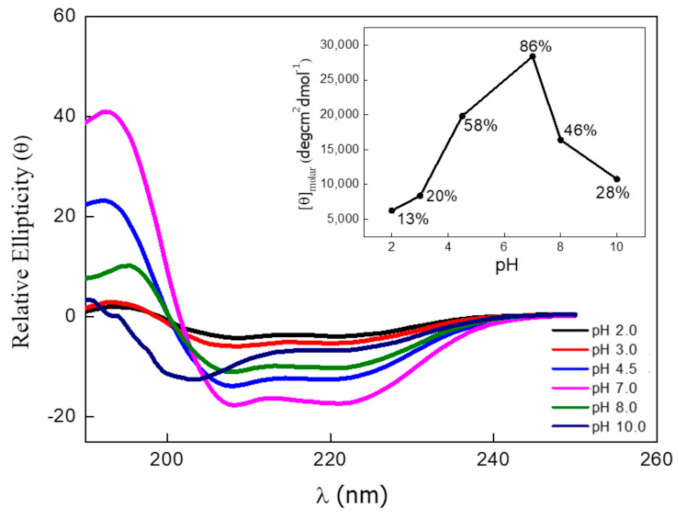
Circular dichroism spectra for AnxA6 (2 µg·mL^−1^) at different pH values. Insertion: Relative molar ellipticity as a function of pH variation with the respective percentage estimates of α-helix structures. Black—pH 2.0 (13%); Red—pH 3.0 (20%); Azul—pH 4.0 (58%); Pink—pH 6.5 (86%); Green—pH 8.0 (46%); and Purple—pH 10.0 (28%).

**Figure 5 ijms-23-08945-f005:**
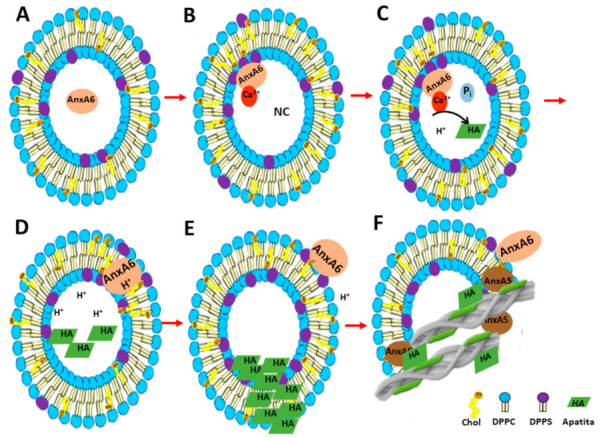
Representation of the action mechanism of annexins during MV-mediated mineralization: (**A**) AnxA6 can be localized in the lumen of MVs at low calcium concentrations (Ca^2+^). AnxA5 can be bound to membrane bilayer; (**B**) Accumulation of Ca^2+^ and inorganic phosphate (Pi) inside MVs favors the binding of AnxA6 to the inner leaflet of the MV membrane bilayer in a Ca^2+^-dependent manner, forming a nucleational core and initiating apatite formation; (**C**) During apatite formation, protons (H^+^) are released, inducing the protonation of AnxA6 and rendering it more hydrophobic due to an ionization potential around 5.5. Protonated AnxA6 can translocate within the MV bilayer; (**D**) Apatite crystals inside the vesicles continue growing up, breaking the membrane of MVs, exposing the HA crystals to the extracellular matrix, and releasing the H^+^ which decreases the local pH; (**E**) AnxA6 can be deprotonated and released to the external surface of MVs. (**F**) AnxA5 present in the membrane bilayer showed a high affinity to collagen binding. Thus, AnxA5 can promotes the adhesion of MVs to collagen fibers, allowing the deposition of the HA crystals on the extracellular matrix.

**Table 1 ijms-23-08945-t001:** Biophysical properties of liposomes and proteoliposomes constituted by different lipids. The diameters were obtained by DLS in triplicate assays, as described in Section 3, and PI is a polydispersion index. “-“ indicate the absence of AnxA6.

Liposome/ Proteoliposome Lipid Composition	AnxA6	Diameter (nm)	PI
**DPPC**	-	132.2 ± 8.0	0.08
	+	130.6 ± 1.7	0.04
**DPPC:DPPS (9:1)**	-	114.4 ± 0.7	0.06
	+	117.9 ± 0.9	0.07
**DPPC:Chol:DPPS (5:4:1)**	-	199.1 ± 1.1	0.08
	+	199.5 ± 1.4	0.09

**Table 2 ijms-23-08945-t002:** The slope of the monotone increase in turbidity and the kinetic parameters of the nucleation process adjusted by the subtraction of monotone process (Appendix A), obtained from the mineralization curves for proteoliposomes and for AnxA6 (control) after 30 h of assay in SCL medium containing PS–CPLX in the absence of type-I collagen. Data are reported as the mean ± SD of triplicate measurements of three different proteoliposome preparations.

Proteoliposome	Parameters					
Lipid Composition	Monotone Slope (U/h)	t_i_ (h)	t_f_ (h)	t_max rate_ (h)	U_max_ (U)	PMP (h^−1^)
**DPPC**	0.0084	9.5 ± 0.54	24.8 ± 0.56	18.5 ± 0.28	0.19 ± 0.05	0.010
**DPPC:DPPS (9:1)**	0.0022	14.9 ± 0.39	24.3 ± 0.18	22.9 ± 0.35	0.15 ± 0.05	0.007
**DPPC:Chol:DPPS (5:4:1)**	0.0013	10.7 ± 0.15	22.5 ± 0.34	19.5 ± 0.55	0.22 ± 0.09	0.011
**Control**	-	20.7 ± 0.41	24.5 ± 0.32	22.4 ± 0.21	0.42 ± 0.03	0.019

**Legend:** t_i_, initial time of mineral formation; t_f_, final time of mineral formation; t_max rate_, time at maximum rate of mineral formation; U_max_, maximum value of turbidity; PMP = U_max_/t_max rate_, potential of mineral propagation.

**Table 3 ijms-23-08945-t003:** Kinetic parameters of the nucleation process obtained from the mineralization curves for proteoliposomes and AnxA6 in solution (control), after 30 h of assay in SCL medium containing PS–CPLX in the presence of type-I collagen. Data are reported as the mean ± SD of triplicate measurements of three different proteoliposome preparations.

Proteoliposome Composition	Parameters				
	t_i_ (h)	t_f_ (h)	t_max rate_ (h)	U_max_ (U)	PMP (h^−1^)
**DPPC**	14.9 ± 0.40	27.5 ± 0.89	20.5 ± 0.63	0.33 ± 0.05	0.016
**DPPC:DPPS (9:1)**	15.2 ± 0.44	29.5 ± 0.57	23.2 ± 0.22	0.39 ± 0.07	0.017
**DPPC:Chol:DPPS(5:4:1)**	13.8 ± 0.28	29.9 ± 0.55	21.7 ± 0.39	0.38 ± 0.02	0.018
**Control**	ND	ND	ND	ND	ND

**Legend:** ND, the curve did not fit; t_i_, initial time of mineral formation; t_f_, final time of mineral formation; t_max rate_, time at maximum rate of mineral formation; U_max_, maximum value of turbidity; PMP = U_max_/t_max rate_, potential of mineral propagation.

**Table 4 ijms-23-08945-t004:** Ratios between the intensity of the bands at 1040 (PO_4_^3−^) and 987 (HPO_3_^2−^) cm^−1^ calculated from the ATR-FTIR spectra (Figure 3) of the mineral precipitated in the in vitro mineralization assay by the AnxA6—containing proteoliposomes and AnxA6 in solution, as a control, in the absence of type-I collagen.

Proteoliposome Composition	PO_4_^3−^/HPO_4_^2−^
DPPC	1.2
DPPC:DPPS (9:1)	0.79
DPPC:Chol:DPPS (5:4:1)	0.64
Control	0.95

## Data Availability

Not applicable.

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
