# Peer review of "Mineralization Profile of Annexin A6-Harbouring Proteoliposomes: Shedding Light on the Role of Annexin A6 on Matrix Vesicle-Mediated Mineralization"

_ijms, 2022, doi:10.3390/ijms23168945_

Round 1
Reviewer 1 Report
The manuscript is well-organized and has relevant results
and interesting information for the readership. However, some minor points should be addressed:
1) I suggest to improve the quality of all the figures adding the legend (colour code).
This will help the reader in the comprehension of all the plots.
2) Explain why a linear increase in the turbidity cannot be ascribed to a nucleation process while a
sigmoid highlights a nucleation step
3) Page 8 insert the Zn2+ in the list of elements that can substitute Ca in the hydroxyapatite
4) Page 8 explain how is estimated the percentage of Ca2+ precipitated
Author Response
Dear Ms. Aisa Safaya
Assistant Editor
Thank you for the opportunity to prepare the revised version of our manuscript entitled “Mineralization profile of annexin A6-harbouring proteoliposomes: shedding light on the role of annexin A6 on matrix vesicle-mediated mineralization” by Ekeveliny Amabile Veschi, Maytê Bolean, Luiz Henrique da Silva Andrilli, Heitor Gobbi Sebinelli; Agnieszka Strzelecka-Kiliszek, Joanna Bandorowicz-Pikula, Slawomir Pikula, Thierry Granjon, Saida Mebarek, David Magne, Ana Paula Ramos, José Luis Millán, Rene Buchet, Massimo Bottini and Pietro Ciancaglini.
The comments helped us to improve the quality of the manuscript. All the questions raised by the reviewers were answered as described below. Also, slight changes were made in the main text. We hope the new version is now suitable for publication in the special issue of IJMS.
Reviewer#1
Comment: I suggest improving the quality of all the figures adding the legend (colour code). This will help the reader in the comprehension of all the plots.
Answer: We thank the reviewer for this suggestion. We have added the symbols and color codes to all figures to make the interpretation clear for the readers.
Comment: Explain why a linear increase in the turbidity cannot be ascribed to a nucleation process while a sigmoid highlights a nucleation step
Answer: The synthetic cartilage medium with 1.41 mM Pi and 2 mM Calcium is metastable and induces mineral precipitation. A linear increase reflects a steady stochastic formation of precipitates, while the nucleation process is a synergic process where the mineral formation increases rapidly, yielding to a characteristic sigmoidal curve with a sharp transition. Sigmoidal curves of the nucleation process in a synthetic cartilage lymph buffer have been reported by Genge et al. [see ref. 22 in the main text] and Wuthier et al. [see ref 32 in the main text]. This comment was added in the manuscript, please see the revised version of the main text.
Comment: Page 8 insert the Zn2+ in the list of elements that can substitute Ca2+ in the hydroxyapatite
Answer: We thank the reviewer for pointing this you. We added Zn2+ as suggested. Please, refer to the revised version.
Comment: Page 8 explain how is estimated the percentage of Ca2+ precipitated
Answer: The concentration of Ca2+ was estimated based on the study by Wu et al. [33]. The authors use the changes in turbidity at 340 nm: an increase of 0.1 in turbidity is equivalent to the precipitation of approximately 10% of the total Ca2+ in synthetic cartilage lymph. Our samples were incubated in the same synthetic cartilage lymph as reported. So, we estimated the percentage of Ca2+ precipitated in the presence of AnxA6 in solution and bound to the proteoliposomes, in the presence of PS-CPLX. We added more information about this in the manuscript, please see the revised version of the main text.
Reviewer 2 Report
Although there is an appreciable effort to use interesting materials and methodologies, results obtained are not enough to accept or refute the main hypothesis. The authors attempted to overcome these shortfalls proposing some new, but poorly supported hypothesis.
Author Response
Dear Ms. Aisa Safaya
Assistant Editor
Thank you for the opportunity to prepare the revised version of our manuscript entitled “Mineralization profile of annexin A6-harbouring proteoliposomes: shedding light on the role of annexin A6 on matrix vesicle-mediated mineralization” by Ekeveliny Amabile Veschi, Maytê Bolean, Luiz Henrique da Silva Andrilli, Heitor Gobbi Sebinelli; Agnieszka Strzelecka-Kiliszek, Joanna Bandorowicz-Pikula, Slawomir Pikula, Thierry Granjon, Saida Mebarek, David Magne, Ana Paula Ramos, José Luis Millán, Rene Buchet, Massimo Bottini and Pietro Ciancaglini.
The comments helped us to improve the quality of the manuscript. All the questions raised by the reviewers were answered as described below. Also, slight changes were made in the main text. We hope the new version is now suitable for publication in the special issue of IJMS.
Reviewer#2
Although there is an appreciable effort to use interesting materials and methodologies, results obtained are not enough to accept or refute the main hypothesis. The authors attempted to overcome these shortfalls proposing some new, but poorly supported hypothesis.
Answer: We thank the reviewer for the comment. We highlighted that the conclusions of the present manuscript were based on previous findings carried out by our group using AnxA6 as well as AnxA5, [12,23,26,28,30,31,34,56,61-63]. Summing up those already reported data with the data presented herein, we could describe the molecular interactions between annexins and different proteins/lipids to recreate biomineralization. We changed the main text to highlight the role of AnxA6 with type-I collagen in the nucleation of the first apatite seeds during biomineralization.
Reviewer 3 Report
The manuscript "Mineralization profile of annexin A6-harboring proteoliposomes: shedding light on the role of annexin A6 on matrix vesicle-mediated mineralization" by Ekeveliny Amabile Veschi et al. is well written and adds a piece to the complex process of bone biomineralization, which still has stages that are not fully elucidated.
However, a few minor corrections/additions need to be made: the first page is missing keywords: please insert them. Also, given the high scientific value of IJMS, the conclusions of the article lack future research perspectives. For example, how the interesting mechanism proposed in the cell-free models will be verified in cells. With what experimental models and methods will it be possible to proceed so that the knowledge gained can have spin-offs in medical fields such as regenerative medicine. In my opinion, this would be important for the scientific community.
Finally, there are some typos: for example, Figure 4 is written in italics and a comma was used in the pH values instead of a period. It has also been corrected mL instead of ml.
Author Response
Dear Ms. Aisa Safaya
Assistant Editor
Thank you for the opportunity to prepare the revised version of our manuscript entitled “Mineralization profile of annexin A6-harbouring proteoliposomes: shedding light on the role of annexin A6 on matrix vesicle-mediated mineralization” by Ekeveliny Amabile Veschi, Maytê Bolean, Luiz Henrique da Silva Andrilli, Heitor Gobbi Sebinelli; Agnieszka Strzelecka-Kiliszek, Joanna Bandorowicz-Pikula, Slawomir Pikula, Thierry Granjon, Saida Mebarek, David Magne, Ana Paula Ramos, José Luis Millán, Rene Buchet, Massimo Bottini and Pietro Ciancaglini.
The comments helped us to improve the quality of the manuscript. All the questions raised by the reviewers were answered as described below. Also, slight changes were made in the main text. We hope the new version is now suitable for publication in the special issue of IJMS.
Reviewer#3
Comment: The manuscript "Mineralization profile of annexin A6-harboring proteoliposomes: shedding light on the role of annexin A6 on matrix vesicle-mediated mineralization" by Ekeveliny Amabile Veschi et al. is well written and adds a piece to the complex process of bone biomineralization, which still has stages that are not fully elucidated.
However, a few minor corrections/additions need to be made: the first page is missing keywords: please insert them. Also, given the high scientific value of IJMS, the conclusions of the article lack future research perspectives. For example, how the interesting mechanism proposed in the cell-free models will be verified in cells. With what experimental models and methods will it be possible to proceed so that the knowledge gained can have spin-offs in medical fields such as regenerative medicine. In my opinion, this would be important for the scientific community.
Answer: We thank the reviewer for all the positive comments about the manuscript and for pointing the typos out. We have revised the main text. The first page and keywords were added. We enlarged the future research perspectives, especially toward possible applications in the regenerative medicine. In the updated manuscript we added: “Membrane models such as proteoliposomes and Langmuir monolayers bearing specific proteins have been useful to investigate the mechanisms of apatite formation and the role of several proteins/enzymes in biomineralization [12,23,26,28,30,31,34,56,61-63]. The use of mimetic models of MVs combined with engineered proteins (AnxA6, AnxA5, TNAP, NPP1, PiT1/2 etc.) could positively impact translational medicine through the development of proteoliposomes as nano reactors of mineral formation for the treatment of bone-related diseases. Moreover, these findings provide information about the mineralization capacity of MVs with molecular details. This will ultimately reinforce the current paradigm regarding the biochemical mechanisms that regulate the role of MVs in physiological and pathological calcification.”
Finally, there are some typos: for example, Figure 4 is written in italics and a comma was used in the pH values instead of a period. It has also been corrected mL instead of ml.
Answer: It was corrected.